# The Association Between Cadmium Exposure and Prostate Cancer: An Updated Systematic Review and Meta-Analysis

**DOI:** 10.3390/ijerph21111532

**Published:** 2024-11-19

**Authors:** Giorgio Firmani, Manuela Chiavarini, Jacopo Dolcini, Stefano Quarta, Marcello Mario D’Errico, Pamela Barbadoro

**Affiliations:** 1Department of Biomedical Sciences and Public Health, Section of Hygiene, Preventive Medicine and Public Health, Polytechnic University of the Marche Region, 60131 Ancona, Italy; g.firmani@pm.univpm.it (G.F.);; 2Department of Health Sciences, University of Florence, Viale GB Morgagni 48, 50134 Florence, Italy

**Keywords:** prostate cancer, cadmium, exposure assessment, meta-analysis, systematic review

## Abstract

Prostate cancer (PCa) is a common cancer among men, and it has a multifactorial etiology. Cadmium (Cd), a toxic heavy metal classified as a carcinogen by the IARC, can cause various acute and chronic effects. This systematic review and meta-analysis aims to update previous findings on the association between Cd exposure and PCa. We carried out a literature search in PubMed, Web of Science, and Scopus up to May 2024, identifying eight new articles. The effect size from the highest and lowest exposure categories were extracted and analyzed using a random-effects model. Heterogeneity was assessed with the I^2^ test, and publication bias was evaluated using funnel plot asymmetry. Overall, the effect size for PCa associated with Cd exposure, combining previous and new articles, was 1.11 (95% CI 0.85–1.45). Heterogeneity was high, but no significant publication bias was detected. For studies from Europe, the effect size was 1.47 (95% CI 1.00–2.17). Notably, 11 articles from the previous systematic review and meta-analysis highlighted that higher Cd exposure is significantly associated with more aggressive histopathological types of PCa (OR 1.50, 95% CI 1.08–2.07). These findings suggest a potential public health concern, indicating the need for further research to clarify the risk associated with Cd exposure.

## 1. Introduction

According to 2020 statistics from the International Agency for Research on Cancer (IARC) GLOBOCAN cancer statistics, PCa is the most common cancer by incidence in men in 118 countries, accounting for one in every fourteen cancers diagnosed globally and 15% of all male cancers [1,2,3]. The etiology of PCa is multifactorial, involving genetic, hormonal, lifestyle, and environmental factors [4]. Among these, the role of environmental and occupational exposures to heavy metals, particularly Cd, has drawn considerable attention due to cadmium’s well-established carcinogenic properties [5].

Cadmium (Cd) is classified as a Group 1 human carcinogen by the IARC, and it has been linked to several types of cancers, including lung, kidney, bladder, and prostate cancers [6,7]. Specifically, Cd exposure has been implicated in the development of more aggressive and advanced stages of prostate cancer (PCa), making it a significant concern for public health. Additionally, the incidence of PCa shows considerable variation across different populations. For instance, African American men have higher rates of PCa incidence and mortality compared to European men, with genetic, environmental, and socio-economic factors playing a role. Understanding the relationship between Cd exposure and these disparities in PCa incidence is crucial for targeted prevention strategies [8].

Cd is a heavy metal with widespread industrial use and environmental presence found in various products, such as batteries, pigments, coatings, and plastics. It can be absorbed by the human organism through various routes, including dietary intake from contaminated food and water, inhalation of cigarette smoke, and occupational exposure in certain industries, such as manufacturing, metallurgy, and battery production [5]. This persistent environmental pollutant is ranked #7 on the 2017 Agency for Toxic Substances and Disease Registry’s Substance Priority List and has been identified as a significant public health concern by the World Health Organization [9,10].

Notably, a Western dietary pattern has been implicated in increasing the risk of PCa. This diet, which is characterized by high consumption of processed meats, refined grains, sweets, and high-fat dairy products, has also been associated with increased Cd intake. In fact, Cd can contaminate various foodstuffs, including cereals, vegetables, and meat products, which are staples in a Western diet [11,12]. This contamination occurs through the uptake of Cd by plants from contaminated soil and water, which is often exacerbated by the use of phosphate fertilizers containing Cd [13].

Recent studies on the link between Cd exposure and PCa incidence have shown mixed results. Some epidemiological research suggests a positive correlation, indicating that higher Cd exposure through diet, environment, and occupation may elevate PCa risk [5,14]. Conversely, other studies have not found significant associations, possibly due to differences in study design, exposure assessment methods, and population characteristics [15,16,17].

Given the critical public health implications and the inconsistent findings across individual studies, a comprehensive systematic review and meta-analysis is necessary to clarify the relationship between Cd exposure and PCa. This systematic review and meta-analysis aim to evaluate the association between Cd exposure from environmental, dietary, and occupational sources and PCa by synthesizing evidence from recent studies to provide a clearer understanding of this potential risk factor. Additionally, to further explore the potential health impacts of Cd exposure, this systematic review and meta-analysis also examines the association between Cd levels and PCa severity. Specifically, it focuses on the more aggressive forms of cancer, which are more dangerous.

## 2. Materials and Methods

This systematic review and meta-analysis adhered to the MOOSE (Meta-Analysis of Observational Studies in Epidemiology) guidelines [18] and the PRISMA statement [19]. The study protocol was registered in the International Prospective Register of Systematic Reviews (www.crd.york.ac.uk/PROSPERO/, registration No: CRD42024541541) (accessed on 13 May 2024).

### 2.1. Search Strategy and Data Source

A systematic search was performed up to 28 May 2024 across the PubMed (http://www.ncbi.nlm.nih.gov/pubmed/ accessed on 1 June 2024), Web of Science (http://wokinfo.com/ accessed on 1 June 2024), and Scopus (https://www.scopus.com/ accessed on 1 June 2024) databases to find original articles related to the association between Cd exposure and PCa. Keywords used included (cadmium) AND (prostate cancer).

The different associations of keywords combined with Boolean operators used for each database are shown in Appendix A.

No publication date limitation was applied, but due to translation restrictions, only English-language studies were eligible.

Additionally, reference lists from included articles and recent relevant reviews were manually checked for further relevant publications.

All results were then screened with Microsoft^®^ Excel^®^ for Microsoft 365 MSO (Version 2409 Build 16.0.18025.20160) 32-bit version.

### 2.2. Eligibility Criteria

Of the selected articles, only those that met the following criteria were included: (i) evaluated the relationship between Cd exposure and PCa; (ii) used a case-control, cohort, or cross-sectional study design; and (iii) reported the odds ratio (OR), relative risk (RR), or hazard ratio (HR), estimated with 95% confidence intervals (CIs).

Studies including aggregated data of PCa cases with other cancer types were excluded. Two investigators independently performed study selection, data extraction, and quality assessment. Disagreements were resolved through discussion or consultation with a third author. Reviews and meta-analyses were excluded. No studies were excluded for weaknesses of design or data quality.

### 2.3. Data Extraction and Quality Assessment

Data extraction was conducted using Microsoft^®^ Excel^®^ for Microsoft 365 MSO (Version 2409 Build 16.0.18025.20160) 32-bit version, and then data were added to a table in Microsoft^®^ Word for Microsoft 365 MSO (Version 2409 Build 16.0.18025.20160) 32-bit version.

Information extracted from each selected study included the first author’s last name, the publication year, the country, the study design, the sample size, population characteristics (age, race, BMI, smoking status), the follow-up duration for cohort studies, risk estimates with 95% CIs, Cd exposure details (biological sample, type of exposure, evaluation test), confirmation of PCa, and adjustment for confounding factors. The Newcastle–Ottawa Scale (NOS) was used to evaluate study quality, with a 9-star system (Appendix A). The full score was 9, and a total score of ≥7 indicated high-quality studies [19]. Two investigators independently carried out the selection of studies, extracted the data, and assessed the quality of the research. Any disagreements were settled through discussion or by involving a third author.

### 2.4. Statistical Analysis

The meta-analysis estimated the overall effect size to assess the association between Cd exposure and PCa. Values from multivariable models accounting for available confounding factors were selected. Due to high heterogeneity, a random-effects model and inverse variance weighting methods were used to compute the combined OR and 95% CIs. Statistical significance was defined as a two-tailed *p*-value < 0.05. The chi-square-based Cochran’s Q statistic and the I^2^ statistic were used to evaluate heterogeneity in results across studies [20]. The I^2^ statistic yield results ranged from 0% to 100% (I^2^ = 0–25%, no heterogeneity; I^2^ = 25–50%, moderate heterogeneity; I^2^ = 50–75%, large heterogeneity; and I^2^ = 75–100%, extreme heterogeneity) [21]. The results of the meta-analysis may be biased if the probability of publication is dependent on the study results. To address potential publication bias, Begg and Mazumdar’s [22] and Egger et al.’s [23] methods were employed to test for funnel plot asymmetry. Sensitivity analyses were carried out to evaluate the impact of individual studies and potential bias on the overall estimate. A funnel plot was considered asymmetrical if Egger’s regression intercept deviated from zero with a *p*-value < 0.05. Analyses were performed using the ProMeta 3 statistical program and STATA 13 for data calculation.

## 3. Results

### 3.1. Study Selection

In this systematic review and meta-analysis, a previous systematic review and meta-analysis that included 22 articles was used as a starting point [24]. Of these, 11 studies specifically addressed the association of PCa in relation to Cd exposure [16,17,25,26,27,28,29,30,31,32,33]. The remaining 11 articles were not considered because they focused on mortality. The current update incorporates 8 additional articles [15,34,35,36,37,38,39,40], bringing the total number of studies analyzed to 19.

Figure 1 shows the details of the research selection process in this study. A total of 238 articles were found through online database searching. After removing duplicated results, 183 studies were evaluated in the title and abstract screening, and 174 records were excluded in this step, based on the following criteria: reviews, pooled studies, meta-analyses, commentaries, and case studies. This exclusion process was vital to ensure that only relevant primary research articles were considered for detailed review and meta-analysis. Then, nine studies were evaluated through full-text review. In addition to database searches, we identified three additional records through reference lists of included articles and recent reviews. These supplementary records [38,39,40] were added to enhance the comprehensiveness of our analysis. A total of 12 reports underwent a detailed eligibility assessment. During this phase, we excluded three reports. Specifically, the studies by [41] and [42] were excluded due to the absence of outcomes. Additionally, the report by [43] was excluded because it contained aggregate data on different types of cancer, which did not align with our specific research focus. Finally, we included nine studies in the systematic review. Of these, eight reports were considered pivotal for the final systematic review and meta-analysis, providing robust data on the incidence outcomes pertinent to our research question [15,34,35,36,37,38,39,40]. One study included in the systematic review was not incorporated into the meta-analysis because it did not provide OR values [44].

### 3.2. Study Characteristics and Quality Assessment

The general characteristics of the nine studies and the relative populations included in the systematic review evaluating the association between Cd exposure and PCa are shown in Table 1. Eight of these studies are included in the systematic review and meta-analysis [36,37,38,39,40,41,42,43].

Of the eight studies included in the systematic review and the meta-analysis, five were case-control studies [35,38,39,40,44], three were cohort studies [15,36,37], and one was a cross-sectional study [34]. There were four studies conducted in America [34,38,39,40], three studies conducted in Europe [15,36,37], and one in Africa [35]. The sample size ranged from 113 to 94,337. Of these studies, Cd exposures were determined from occupational, dietary, and environmental matrices.

Some studies utilized blood samples as the biological specimen [34,35,36,40,44], while others used urine samples [35,40]. Nyqvist et al. assessed Cd exposure through environmental and occupational sources, while Eriksen et al. evaluated Cd exposure through dietary intake using a food frequency questionnaire (FFQ). Finally, Aronson et al. and Elghany et al. assessed Cd exposure through occupational history and histologically confirmed PCa. The health outcome investigated was the association between Cd exposure and PCa. The study-specific quality scores of the selected studies are shown in the last column on the right of Table 1, which ranged from 6 to 9 (median: 7; mean: 7).

From the previous systematic review and meta-analysis, we extracted only the articles that reported OR and RR and investigated the association between Cd exposure and PCa. The general characteristics of the 11 studies and the relative populations included in the previous systematic review and meta-analysis evaluating the association between Cd exposure and PCa are shown in Table 1.

Of the eleven studies included in the systematic review and the meta-analysis, nine were case-control studies [26,27,28,29,30,31,32,33] and two were cohort studies [17,25]. There were three studies carried out in America [16,31,32], six studies performed in Europe [25,27,28,29,30,33], and two in Asia [17,26]. The sample size ranged from 39 to 46,003. Of these studies, Cd exposures were determined from occupational, dietary, and environmental exposure.

Among the eleven studies analyzed, blood and urine were used as biological samples in one study (specifically, Chen et al.). This study examined 261 cases and 267 controls using both blood and urine to measure Cd exposure. Platz et al., who conducted a USA nested case-control study with 115 cases and 227 controls, used toenail samples to evaluate Cd levels, as in the case-control studies by Vinceti et al. and Seidler et al. The remaining studies did not specify the biological samples used. Armstrong and Kazantzis assessed exposure using occupational history. Checkoway et al. conducted a study with 40 cases and 40 controls utilizing a structured occupational questionnaire, which was similar to the approach taken by van der Gulden et al. West et al. used a food frequency questionnaire (FFQ) to assess exposure, which was similar to the Swedish study by Julin et al. and the Japanese study by Sawada et al., which were both cohort studies. Finally, Rooney et al. based their exposure assessment on occupational health and radiation data. The health outcome investigated was the PCa. Study-specific quality scores, evaluated using the Newcastle–Ottawa Scale (NOS), are shown in the last column on the right of Table 1; they ranged from 3 to 6 (median: 5; mean: 5).

### 3.3. Meta-Analysis

A comprehensive systematic review and meta-analysis was performed to investigate the association between Cd exposure and PCa (Table 2; Figure 2, Figure 3). As shown in Figure 2 for the combined analysis of the 19 articles, the odds ratio OR is 1.11 (95% C.I. 0.85–1.45). This result is not statistically significant, indicating that the association between Cd and PCa is not definitively established. The heterogeneity values are quite high, with Q = 400.16 and I^2^ = 95.50%, suggesting considerable variability among the studies. In the subset of 11 articles, the OR is 1.29 (95% CI 0.75–2.21), which also is not statistically significant for the risk of developing PCa. The heterogeneity remains high, with Q = 290.74 and I^2^ = 96.56%, pointing to substantial differences between the studies. For the eight new articles, the OR is 0.95 (95% CI 0.77–1.16), which is, again, not statistically significant for the risk of developing PCa. The heterogeneity is lower than in the larger sets (19 articles and 11 articles) but still notable, with Q = 34.13 and I^2^ = 79.49%. Focusing on age-adjusted results, none of the 19 articles, including both the previous 11 and the new 8, resulted in a statistically significant risk of developing PCa.

When examining the type of biological sample, we analyzed the relationship between Cd concentrations in blood and urine and their association with PCa risk. Specifically, we assessed how Cd levels in these samples correlate with the risk of developing PCa. Data from 19 studies showed that blood Cd levels presented an OR of 0.75 (95% CI: 0.51–1.10), while urine Cd levels showed an OR of 0.74 (95% CI: 0.56–0.98). The latter is statistically significant, suggesting a potential protective association between Cd in urine and PCa. The heterogeneity for blood is Q = 28.35 and I^2^ = 85.89%, and, for urine, it is Q = 0.72 and I^2^ = 0.00%, indicating no heterogeneity.

For the 11 articles, blood had an OR of 1.44 (95% CI: 0.78–2.64), and urine had an OR of 0.49 (95% CI: 0.31–0.78), with the urine result being statistically significant. This suggests a potential protective association between Cd in urine and PCa, confirming the results obtained for the 19 articles. In the eight articles, blood samples had an OR of 0.75 (95% CI: 0.51–1.10) and urine samples had an OR of 0.76 (95% CI: 0.48–1.20); neither was significant.

Regarding the type of study, for 19 articles, cohort studies show an overall OR of 1.10 (95% CI: 1.00–1.20), which is marginally significant, suggesting a slight association with the risk of developing PCa. The heterogeneity is Q = 9.27 and I^2^ = 56.85%. Case-control studies show an increased likelihood of PCa risk with an OR of 1.17 (95% CI: 0.64–2.14), which is not statistically significant, with a high heterogeneity (Q = 186.37 and I^2^ = 93.56%). Regarding the subset including the latter 11 articles, cohort studies have an increased likelihood of PCa risk with an OR of 1.13 (95% CI: 1.03–1.23), which is statistically significant. This result showed no heterogeneity (Q = 0.07 and I^2^ = 0.00%). On the other hand, case-control studies of the same subset of 11 articles did not show a statistically significant increased likelihood of PCa or Cd exposure. In the eight articles, cohort studies show an OR of 1.10 (95% CI: 0.94–1.28), which is not significant, with Q = 8.12 and I^2^ = 75.38%, while case-control studies have an OR of 0.91 (95% CI: 0.64–1.29), which is also not significant for the risk of developing PCa, with Q = 2.73 and I^2^ = 0.00%.

As shown in Table 2, geographically, for 19 articles, European studies show an OR of 1.47 (95% CI: 1.00–2.17), which is marginally significant for the risk of developing PCa, with Q = 349.22 and I^2^ = 97.71%. Based on an analysis of the type of exposure across the articles, none of the results are statistically significant. This lack of significance is evident in the forest plots in Figure 3 where dietary, occupational, and environmental exposures all show no significant association with PCa.

Regarding cancer severity, which focuses exclusively on the more aggressive forms of cancer, for the analysis of all 19 included articles, the OR is 1.23, with a 95% CI from 0.87 to 1.73. The heterogeneity test shows Q = 24.70 and I^2^ = 91.90%, indicating a high level of variability among the studies, with a *p*-value of 0.00 confirming the significance of this heterogeneity. When the analysis is narrowed down to 11 articles, the OR is 1.50, with a CI of 1.08 to 2.07, suggesting a stronger and statistically significant association with the risk of developing PCa.

Finally, it was not possible to create an association between Cd exposure and PCa by dividing the studies by NOS score due to a lack of statistical significance. Considering the NOS score for quality assessment, for 19 articles, studies with a score of ≥7 show an OR of 1.04 (95% CI: 0.97–1.11), which is not significant, with Q = 5.15 and I^2^ = 2.85%. Studies with a score of <7 have an OR of 1.26 (95% CI: 0.90–1.76), which is not significant, with Q = 344.05 and I^2^ = 96.51%. For 11 articles, the OR is 1.29 (95% CI: 0.75–2.21), which is not significant, with Q = 290.74 and I^2^ = 96.56%. In the eight articles, studies with a score of ≥7 have an OR of 1.04 (95% CI: 0.97–1.11), which is not significant, with Q = 5.15 and I^2^ = 2.85%. Studies with a score of <7 have an OR of 0.86 (95% CI: 0.31–2.40), which is not significant, with Q = 28.01 and I^2^ = 96.43%.

### 3.4. Sensitivity Analysis

Figure 4 presents the funnel plots for the systematic review and the meta-analysis. These plots were utilized to investigate the presence of publication bias and to assess the stability of the overall results. A symmetrical funnel plot indicates the absence of publication bias, while an asymmetrical shape may suggest its presence. Outlier articles were identified and removed to determine if this would improve the results. The underlying assumption was that outliers might distort the overall results, and their exclusion could provide a more accurate and reliable effects size. However, after removing the outlier articles, there was no significant improvement in the results. The sensitivity analysis, which included examining the funnel plots and removing outlier articles, did not show any improvement in the results.

Considering other subgroups, the investigation of the effect a single study may have on the value of PCa indicated that estimates were, in some cases, influenced by a single study. Removal of the study by Vinceti et al. [28] increased the values, leading to a rise in the overall values in the European subgroup, both when considering items from the previous systematic review and meta-analysis and the new one (1.11; 95% CI 1.01–1.22; *p* = 0.033) and when only considering those from the previous one (1.14; 95% CI 1.04–1.25; *p* = 0.005), as shown in Appendix A.

### 3.5. Publication Bias

Publication bias was detected for case-control studies (19 articles and 11 articles) with Egger’s method and for articles with NOS score < 7 (19 articles) with Begg’s method (Table 2; Figure 4).

## 4. Discussion

By comparing the results of the previous systematic review and meta-analysis [24] with those of the current one, this study aims to update and refine the earlier findings. This updated systematic review and meta-analysis not only incorporates the most recent data but also re-evaluates the outcomes of previous findings. Through this comprehensive comparison, several significant points have emerged, confirming some of the earlier results while revealing new evidence. These insights provide a more robust and current understanding of the subject, offering valuable implications for future research and practical applications.

According to the previous systematic review and meta-analysis, we observed that the overall effect size for the association between Cd exposure and PCa is consistent. The OR for all 19 articles, which includes both the 11 articles from the previous systematic review and meta-analysis and 8 new articles, is 1.11 (95% CI 0.85–1.45). This result indicates that there is no significant association between Cd exposure and PCa, which is in agreement with the previous systematic review and meta-analysis (OR = 1.23, 95% CI 0.81–1.88). For the 11 articles included in the previous systematic review and meta-analysis, the OR is 1.29 (95% CI 0.75–2.21), and for the 8 newly selected articles, the OR is 0.95 (95% CI 0.77–1.16). Notably, the confidence interval for the 8 new articles (0.77–1.16) is significantly reduced compared to that of the 11 articles from the previous systematic review and meta-analysis (0.75–2.21). In fact, the width of the confidence interval is reduced by 1.07 units. This reduction indicates a greater precision in the estimates provided by the new studies. These results suggest that the inclusion of the new studies did not alter the overall conclusion of the previous analysis: there is no significant association between Cd exposure and PCa. Thus, the findings of the updated systematic review and meta-analysis reinforce the earlier conclusion that Cd levels are not significantly associated with PCa considering the overall effect.

An important aspect of the updated systematic review and meta-analysis is the stratification by biological sample type, which was not performed in the earlier study. The updated analysis, considering 19 articles, reveals that increased Cd presence in urine is negatively associated with developing PCa, with an OR of 0.74 (95% CI 0.56–0.98). This trend is also confirmed by analyzing the 11 articles in the previous systematic review and meta-analysis (OR = 0.49, 95% CI 0.31–0.78). This finding may suggest that Cd could increase PCa risk predominantly through bloodstream exposure rather than the urinary tract, which may depend on the biological mechanism involved. Cd is primarily absorbed through the respiratory and gastrointestinal tracts, accumulating mainly in the kidneys and liver. The clearance of Cd from the body occurs mainly through urine, but it is a slow process due to the long biological half-life of Cd in tissues (6–38 years in the kidneys and 4–19 years in the liver) [45], with a possible continuous and long-term period of transfer from storage organs to the bloodstream and vice versa. Increased urinary excretion of Cd can reduce its blood concentration, thus lowering cancer risk incidence. In any case, this is only a hypothesis that requires further study, as we cannot exclude that increased Cd excretion can be a sign of kidney damage caused by long-term Cd exposure, which could be a sign of systemically increased and not decreased risk.

Moreover, the earlier systematic review and meta-analysis did not find significant associations for cohort studies. However, in the updated systematic review and meta-analysis considering 11 articles from the previous systematic review and meta-analysis, cohort studies show a significant association with an OR of 1.13 (95% CI 1.03–1.23). This difference might be explained by the fact that the earlier systematic review and meta-analysis considered both incidence and mortality data, whereas the updated analysis focuses solely on incidence. By selecting only studies that reported incidence, the updated systematic review and meta-analysis narrow the scope and potentially refine the results.

Additionally, the updated systematic review and meta-analysis find that living in Europe, based on 19 studies, is marginally significantly associated with higher Cd exposure and with developing PCa, with an OR of 1.47 (95% CI 1.00–2.17), probably due to higher Cd exposure. This result is consistent with reports indicating higher Cd levels in European environments [46]. In fact, Nawrot et al. established that European environments often have elevated Cd levels due to industrial activities, agricultural practices, and contaminated water sources, highlighting the increased cancer risk associated with such exposures. Moreover, elevated Cd exposure in Europe is influenced not only by environmental factors but also by lifestyle choices, including diet [47]. Specifically, a Western dietary pattern, which often includes foods contaminated by Cd, contributes to this increased exposure. Fabiani et al. demonstrated that a Western dietary pattern increases PCa risk, thus reflecting the impact of diet on Cd exposure and cancer risk [12]. Nonetheless, it is important to highlight that although we found positive findings in European studies, we did not perform a dose–response analysis, as this was beyond the scope of the work. Therefore, we can only hypothesize that Europeans may have higher cadmium (Cd) exposure, although we lack direct evidence. Additionally, other factors beyond dietary patterns should be considered with further investigation.

Finally, the updated systematic review and meta-analysis offers new insights by addressing cancer severity, a factor that was not considered in the previous analysis. Specifically, this new systematic review and meta-analysis shifts its focus to the most aggressive forms of PCa. This represents a significant advancement from the previous analysis, which did not differentiate between the severity of PCa cases. The updated systematic review and meta-analysis highlights that higher Cd exposure is significantly associated with the development of aggressive PCa. The inclusion of cancer severity in the analysis underscores the need for targeted preventive measures and interventions. Understanding that Cd exposure is linked to more aggressive PCa forms, it emphasizes the importance of reducing exposure to this toxic metal to mitigate its impact on health outcomes. It also calls for further research into the mechanisms through which Cd contributes to cancer severity.

Furthermore, while the earlier systematic review and meta-analysis used a Newcastle–Ottawa Scale (NOS) score cutoff of 4.8 to differentiate study quality, the updated analysis appropriately uses a cutoff of 7. This distinction did not reveal any significant statistical differences, thus mirroring the findings of the earlier study.

The present systematic review and meta-analysis has several strengths. Unlike the previous systematic review and meta-analysis on Cd exposure and risk of PCa [24], which did not distinguish between incidence and mortality, we were able to analyze and consider the association between Cd exposure and PCa, excluding mortality. In addition, our study showed the association between Cd and PCa in several different geographic areas. Moreover, we have been able to stratify for important factors, such as biological samples and cancer severity. In the end, we did not observe significant publication bias in the studies selected.

On the other hand, our study has some limitations that need to be discussed. Due to the relatively small number of studies included, our systematic review and meta-analysis may restrict the statistical power to evidence related to association, thus preventing the results’ generalization on a large scale. This is more evident in the stratified analysis on the same important parameters including type of Cd exposure (environmental, occupational, and dietary). Another important limitation is the possible presence, in studies selected, of an exposure assessment bias. In fact, not all articles used an adequate biological sample, such as blood or urine, for Cd exposure evaluation. Some of them used questionnaires to assess the exposure levels, which is an indirect method of evaluation that can provide imprecise estimates of actual exposure.

## 5. Conclusions

In conclusion, while the updated systematic review and meta-analysis reinforces the findings of the previous study, it also provides new insights through further stratification analyses and considerations. Specifically, this updated systematic review and meta-analysis integrates recent studies that reinforce the actual evidence but also promote future experimental research. A critical focus of this updated systematic review and meta-analysis is on disease severity, where findings suggest a strong association between Cd exposure and the development of more aggressive forms of PCa. These insights represent an important strength of our work, and, if validated by future research, they could be useful in developing targeted public health interventions aimed at reducing Cd exposure. Further studies are necessary to explore these associations more thoroughly.

## Figures and Tables

**Figure 1 ijerph-21-01532-f001:**
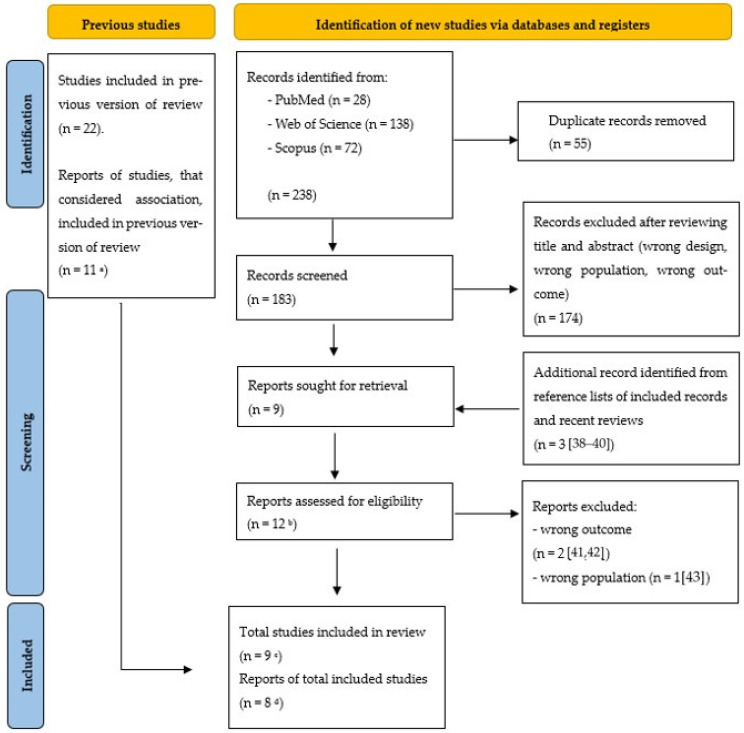
Flow diagram of the systematic literature search on cadmium (Cd) exposure and prostate cancer (PCa). ^a^ [16,17,25,26,27,28,29,30,31,32,33]; ^b^ [15,34,35,36,37,38,39,40,41,42,43,44]; ^c^ [15,34,35,36,37,38,39,40,44]; ^d^ [15,34,35,36,37,38,39,40]. From Page MJ, McKenzie JE, Bossuyt PM, Boutron I, Hoffmann TC, Mulrow CD, et al. The PRISMA 2020 statement: an updated guideline for reporting systematic reviews. BMJ 2021; 372: n71. doi:10.113.6/bmj.n7.

**Figure 2 ijerph-21-01532-f002:**
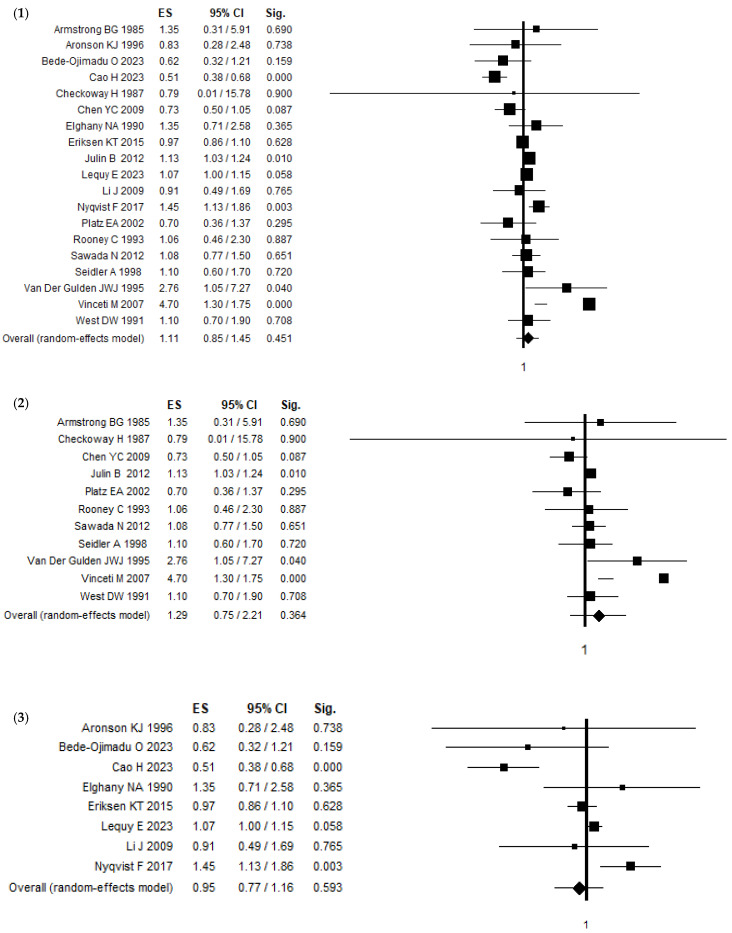
Forest plot of the association between Cd exposure and PCa considering (**1**) all articles [15,16,17,25,26,27,28,29,30,31,32,33,34,35,36,37,38,39,40], (**2**) only articles [16,17,25,26,27,28,29,30,31,32,33] from the previous systematic review and meta-analysis [24], and (**3**) only new ones [15,34,35,36,37,38,39,40].

**Figure 3 ijerph-21-01532-f003:**
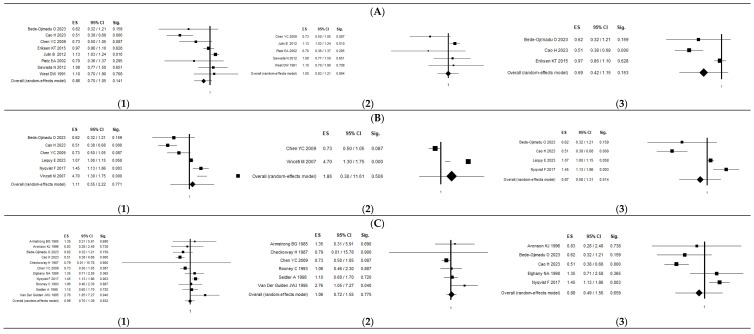
Forest plot of the association between Cd exposure and PCa considering type of exposure: (**A**) dietary, (**B**) environmental, and (**C**) occupational. For each type of exposure, the forest plots are reported by (**1**) all articles [15,16,17,25,26,27,28,29,30,31,32,33,34,35,36,37,38,39,40], (**2**) only articles [16,17,25,26,27,28,29,30,31,32,33] from the previous systematic review and meta-analysis [24], and (**3**) only new ones [15,34,35,36,37,38,39,40].

**Figure 4 ijerph-21-01532-f004:**
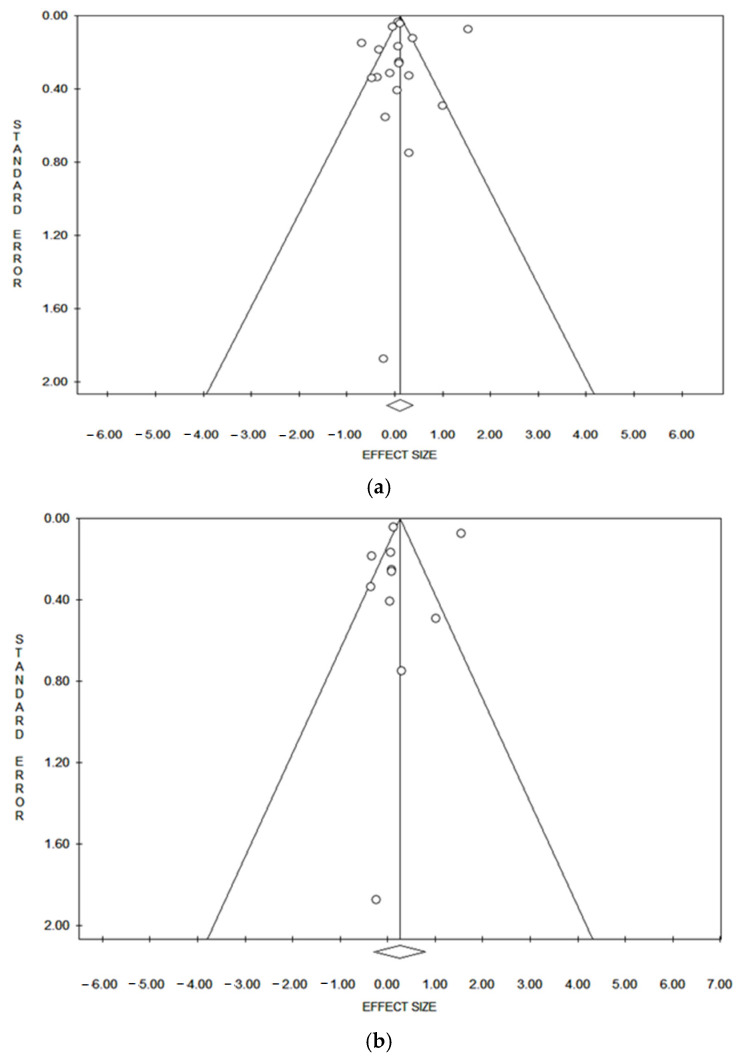
Funnel plot of publication bias of the association between Cd exposure and PCa. (**a**) The 19 articles from a previous systematic review and meta-analysis [24] and the new review, (**b**) 11 articles from a previous systematic review and meta-analysis, (**c**) 8 articles from the new one.

**Table 1 ijerph-21-01532-t001:** Characteristics of the studies included in the systematic review and meta-analysis of the association between Cd exposure and PCa (in bold are the studies extracted from the previous review by Ju-Kun et al., 2016 [24]).

First Author, Year, Location Reference	Study Design C: Cohort CC: Case-Control FU: Follow-Up	Population and Characteristics. N: Number [Age; Smoking (% Never); Race/Ethnicity (%); BMI (kg/m^2^)] M: Mean; Mdn: Median	Cd Exposure (Biological Sample, Type of Exposure, Evaluation Test)	PCa	Results	OR/RR/HR (95% CI)	P for Trend	Matched or Adjusted Variables	NOS
Cao H, 2023 USA [34]	Cross-sectional study from NHANES	N = 94.337 N with PCa = 784 -Age: 71 (M); -Non-Hispanic White: 75.6%; -BMI (kg/m^2^): 28.5 (M)	-Blood -Environmental, occupational, and dietary -ICP-DRC-MS.	NA	Used statistics methods: REF: lowest (lowest vs. highest) WQS SVYGLM	OR 0.52 (0.36–0.76) 0.49 (0.30–0.80)		Age, education, race/ethnicity, poverty income ratio (PIR), BMI, and marital status	7
Bede-Ojimadu O, 2023 Nigeria [35]	CC	N = 273 N with PCa = 82 Controls: 98 -Age: Cases 70.5 (M), Controls 68.5 (M); -BMI (kg/m^2^): Cases 23.66 (M), Controls 24.78 (M); -Never smokers: Cases 39.02%, Controls 28.8%	-Blood, urine -Environmental, occupational, and dietary -ICAP-RQ-ICP-MS.	Histologically confirmed incident PCa	Urinary Cd Q1 (<1.5179) Q2 (1.5180–2.4108) Q3 (2.4109–4.3082) Q4 (>4.3082) Blood Cd Q1 (≤0.8036) Q2 (0.8037–1.7858) Q3 (1.7859–2.5894) Q4 (>2.5894)	OR REF 0.6 (0.23–1.55) 0.73 (0.29–1.83) 0.76 (0.28–2.06) REF 0.77 (0.32–1.83) 0.51 (0.20–1.32) 0.52 (0.21–1.29)		Age, BMI, major occupation, smoking frequency and family history of PCa, residential proximity to known sources of trace element pollution, and physician-diagnosed diabetes	7
Lequy E, 2023 France [36]	C, Gazel Cohort, FU: 1989–2015	N = 11.755 N with PCa = 957 Characteristics of the whole population: -Age: Cases 51.5 (Mdn), Controls 50.5 (Mdn); -BMI (kg/m^2^): Cases 36.7%, Controls 33.5%; -Never smokers: Cases 39.5%, Controls: 42.2%	-Blood -Environmental -ICP-MS.	Incident cancer sources: -French national health administrative databases (81%) OTHER: -Company records for employees -Self-reported by participants	REF: low per IQR	HR 1.07 (1.00–1.15)		Time-varying cumulative smoking pack-years, alcohol use, marital status, fruit and vegetable intake, calendar year; sex, socio-economic status, and education as static variables	7
Lim JT, 2019 Singapore [44]	CC, SPCS	Cases = 141, Controls = 114 -Age (≥70): Cases 35.5%, Controls 22.8% -BMI (≥28 kg/m^2^) Cases 14.2%, Controls 13.2% -Ethnicity (Chinese): Cases 85.1%, Controls 79.8%	-Blood -Environmental -ICP-MS.	Biopsy or operative specimens by pathologist	NA	NA	NA	NA	
Nyqvist F, 2017 Sweden [37]	C study, FU: 1979–2010	N = 34.266 N with PCa = 572	-NA -Environmental, occupational -NA.	ICD-7 177.1	Low Medium High	OR REF 1.05 (0.87–1.27) 1.45 (1.13–1.86)		Age and calendar year	6
Eriksen KT, 2015 Denmark [15]	C study, DCH cohort, FU	N = 26.778 N with PCa = 1.567 Total Cohort characteristics: -Age: 56 (Mdn); -BMI (kg/m^2^): 26 (Mdn); -Never smokers: 26% Cases characteristics: -Age: 58 (Mdn); -BMI (kg/m^2^): 26 (Mdn); -Never smokers: 27%	-NA -Dietary -FFQ.	Danish Cancer Registry	Low (<14 µg/day) Medium (14–18 µg/day) High (>18 µg/day) Cancer severity REF: low Aggressive	IRR REF 0.96 (0.85–1.08) 0.97 (0.86–1.10) 1.00 (0.86–1.16)		Educational level, smoking status, BMI, waist-to-hip ratio, and physical activity	9
Li J, 2009 USA [40]	CC, study period 1994–1996	Cases= 113 Controls = 258 -Age: Cases 66.1 (M), Controls 67 (M); -Race (White): Cases 84.1% Controls 85.3% -BMI (kg/m^2^): Cases 27 (M), Controls 27.3 (M)	-Blood, urine -NA -CS AAS.	Hospital tumor registries	Urinary Cd/creatinine (µg/mg × 10^−4^) <5.34 × 10^−4^ ≥5.34 × 10^−4^	OR REF 0.91 (0.49–1.69)		Urinary creatinine-standardized Cd, age, race, education, smoking status, family history of PC in a first-degree relative, physical activity, vasectomy, multivitamin taking, total energy intake, BMI, STD, and Cd through STD interaction	8
Aronson KJ, 1995 Canada [39]	CC, study period 1979–1986	Cases = 449 -Controls: (1) Populations 533 (2) Other cancer 1550 -Age: Cases 63 (M), Control (1) 57 (M) Control (2) 59 (M)	-NA -Occupational -Job history information that was assessed by exposure experts.	Histologically confirmed	REF: low Non-substantial exposure	OR 0.83 (0.28–2.48)		Age, ethnicity, socioeconomic status, Quetelet index, and self/proxy status of the respondent	9
Elghany NA, 1990 USA [38]	CC	Cases = 358 Controls = 679 -Age (>67): Cases 50% Controls 43.3% -Never smokers: Cases 42.7% Controls 42.4%	-NA -Occupational -Rapid reporting system set up in all of Wasatch Front pathology laboratory.	-Histologically confirmed (code 185.9 by International Classification of Disease 9th revision) -Utah Cancer Registry	REF: low All tumors ^1^ Aggressive tumors ^2^	OR 1.3 (0.6–2.7) 1.5 (0.4–5.1)		Age	7
**Armstrong BG, 1985 United Kingdom** [33]	CC	N of cases: 39; N of controls: 115	-NA -Occupational by work history information -NA.	ICD 185	REF: lowest High	OR 1.35 (0.31–5.91)		Unadjusted	3
**Checkoway H, 1987 USA** [32]	CC, study period: 1984–1985	N of cases: 40; N of controls: 40; Age: -Cases 68.8 (M), -Controls 67.3 (M) Race (White): -Cases 55%, -Controls 73.4%	-NA -Occupational -Structured questionnaire.	-Histologically confirmed PCa	REF: low	OR 0.79 (0.01–15.78)		Unadjusted	3
**West DW, 1991 USA** [31]	CC, study period: 1984–1985	N of cases: 358; N of controls: 679; Age (68–74 years): -Cases 50%, -Controls 43% Smoking status: -Cases 43%, -Controls 42% BMI (≥30 kg/m^2^): -Cases 26%, -Controls 26%	-NA -Dietary -FFQ.	-Histologically confirmed PCa -Utah Cancer Registry -SEER	REF low (<36 µg) Cd intake >61 µg	OR 1.1(0.7–1.9)		Unadjusted	5
**Rooney C, 1993 United Kingdom** [30]	CC, study period: 1946–1986	N of cases: 136; N of controls: 404	-NA -Occupational by occupational health, personnel, and radiation records held by the UKAEA -NA.	-Histologically confirmed PCa -Hospital records	REF: low Work exposure	RR 1.06 (0.46–2.30)		Unadjusted	4
**Van Der Gulden JWJ, 1995 Netherlands** [29]	CC, study period: 1988–1990	N of cases: 345; N of controls: 1346 Age (Mdn): -Cases 72, -Controls 69	-NA -Occupational by questionnaire -NA.	Histologically confirmed PCa	REF: low Cd exposure	OR 2.76 (1.05–7.27)		Age	4
**Seidler A, 1998 Germany** [28]	CC	N of cases: 192; N of controls: 210 Age (Mdn): -Cases 71.1, -Controls 69.7	-Toenail -Occupational -Furnace atomic absorption with Zeeman background correction.	Histologically confirmed PCa	REF: never Low probability of exposure (≥5 years)	OR 1.1 (0.6–1.7)		Age, smoking, and region	4
**Platz EA, 2002 USA** [16]	Nested CC, study period: 1989–1996	N of cases: 115; N of controls: 227	-Toenail -Dietary -Perkin Elmer Model 5100 PC with Zeeman background correction.	Cases were pathologically confirmed through the following: -Washington County Cancer Registry -Maryland Cancer Registry	REF: low (Q1) Q5	OR 0.70 (0.36–1.37)	0.9	Toenail weight using residual analysis	5
**Vinceti M, 2007 Italy** [27]	CC	N of cases: 45; N of controls: 68	-Toenail -Environmental -Perkin Elmer Model 5100 PC with Zeeman background correction.	Histologically based diagnosis of PC	REF: low (Q1) Q4	OR 4.70 (1.30–1.75)	0.004	Unadjusted	5
**Chen YC, 2009 Taiwan** [26]	CC	N of cases: 261; N of controls: 267 Age (Mdn): -Cases 72.1, -Controls 71.3	-Urinary, blood -Environmental, occupational, and dietary -Perkin Elmer Model 5100 PC with Zeeman background correction.	Histologically based diagnosis of PC from four medical centers	REF: low Blood µg/L >0.87 Urinary (µg CD/g creatinine) >1.12 Cancer severity REF: low (GS 2–6) Urine GS > 8 Blood GS > 8	OR 1.44 (0.78–2.64) 0.49 (0.31–0.78) 2.89 (1.25–6.70) 1.58 (1.40–1.77)		Age, smoking, and medical institution	6
**Julin B, 2012 Sweden** [25]	C, FU: 1998–2009	C size: 41.089 N of cases: 3085 Age (Mdn): 56.2	-NA -Dietary -FFQ.	National Cancer Registry	REF: low (<17 µg per day) >20 µg per day Cancer severity REF: low Fatal	RR 1.13 (1.03–1.24) 1.14 (0.86–1.51)		Age, family history of PC, years of education, BMI, waist circumference, metabolic equivalent hours per day, smoking status, total energy intake, alcohol consumption, selenium, lycopene, and calcium	6
**Sawada N, 2012 Japan** [17]	C, FU: 1990–1998	C size: 46.033 N of cases: 470	-NA -Dietary -FFQ.	-Notification from the major hospitals -Population-based cancer registries -Responses to questionnaires	REF: low (19.7 µg per day) High µg per day	HR 1.08 (0.77–1.50)		Age, area, BMI, smoking status, alcohol intake, sports in leisure time, intake of meat, soybean, vegetables, and fruit	6

NA, not available; SPCS, Singapore Prostate Cancer Study; NHANES, National Health and Nutrition Examination Survey; ICP-DRC-MS, Inductively Coupled Plasma Dynamic Reaction Cell Mass Spectrometry; WQS, Weighted Quantile Sums; SVYGLM, Survey Generalized Linear Model; ICP-MS, Inductively Coupled Plasma Mass Spectrometry; SEER, Surveillance, Epidemiology, and End Results; DCH, Danish, Diet, Cancer and Health; STD, sexually transmitted disease.

**Table 2 ijerph-21-01532-t002:** Results of the stratified analysis of the PCa risk estimates according to Cd exposure.

	Combined Risk Estimate (OR)		Test of Heterogeneity			Publication Bias	
	No. ^b^	Value (95% CI)	Q	I^2^ %	*p*	*p* (Egger Test)	*p* (Begg Test)
ALL (19 articles)	23	1.11 (0.85–1.45)	400.16	95.50	0.00	0.893	0.100
ALL (11 articles)	12	1.29 (0.75–2.21)	290.74	96.56	0.00	0.800	0.073
ALL (8 articles)	11	0.95 (0.77–1.16)	34.13	79.49	0.00	0.481	0.621
Type of variable							
Adj for age (19 articles)	16	1.00 (0.85–1.17)	42.60	74.18	0.00	0.542	1.000
Adj for age (11 articles)	6	1.07 (0.85–1.34)	8.70	54.03	0.07	0.961	0.624
Adj for age (8 articles)	10	0.92 (0.68–1.26)	32.92	81.77	0.00	0.525	0.453
Type of biological sample							
19 articles							
Blood	5	0.82 (0.49–1.36)	26.12	88.51	0.00	0.466	0.497
Urine	3	0.65 (0.43–0.98)	2.63	23.97	0.268	0.570	0.602
11 articles							
Blood	1	1.44 (0.78–2.64)	/	/	/	/	/
Urine	1	0.49 (0.31–0.78)	/	/	/	/	/
8 articles							
Blood	4	0.69 (0.37–1.30)	24.94	91.98	0.00	0.372	0.602
Urine	2	0.76 (0.48–1.20)	1.00	0.00	0.608	0.489	0.602
Type of study							
19 articles							
Cohort	5	1.10 (1.00–1.20)	9.27	56.85	0.055	0.562	0.624
Case-control	16	1.17 (0.64–2.14)	186.37	93.56	0.00	0.005	0.143
11 articles							
Cohort	2	1.13 (1.03–1.23)	0.07	0.00	0.798	/	/
Case-control	10	1.33 (0.62–2.84)	147.67	94.58	0.00	0.049	0.211
8 articles							
Cohort	3	1.10 (0.94–1.28)	8.12	75.38	0.02	0.651	0.602
Case-control	6	0.91 (0.64–1.29)	2.73	0.00	0.435	0.833	0.497
Geographic location							
19 articles							
European	9	1.47 (1.00–2.17)	349.22	97.71	0.00	0.545	0.095
United States	9	0.82 (0.58–1.16)	12.25	51.02	0.06	0.241	0.881
Asia	3	0.89 (0.60–1.32)	2.47	59.44	0.12	/	/
Africa	2	0.62 (0.32–1.21)	0.30	0.00	0.581	/	/
11 articles							
European	6	1.71 (0.76–3.85)	259.10	98.07	0.00	0.811	0.348
United States	3	0.93 (0.63–1.39)	1.14	0.00	0.567	0.796	0.602
Asia	3	0.89 (0.60–1.32)	2.47	59.44	0.12	/	/
8 articles							
European	3	1.10 (0.94–1.28)	8.12	75.38	0.02	0.651	0.602
United States	6	0.80 (0.48–1.34)	8.89	66.0	0.032	0.213	0.497
Africa	2	0.62 (0.32–1.21)	0.30	0.00	0.581	/	/
Type of exposure							
19 articles							
Dietary	11	0.86 (0.70–1.05)	33.07	78.83	0.00	0.108	0.322
Occupational	15	0.99 (0.70–1.39)	38.10	73.76	0.00	0.776	0.815
Environmental	9	1.11 (0.55–2.22)	370.3	98.65	0.00	0.995	0.851
11 articles							
Dietary	6	1.00 (0.83–1.21)	6.97	42.62	0.137	0.171	0.327
Occupational	7	1.06 (0.72–1.55)	7.33	31.82	0.197	0.261	0.851
Environmental	3	1.86 (0.30–11.61)	85.18	98.83	0.00	/	/
8 articles							
Dietary	5	0.69 (0.42–1.15)	16.52	87.89	0.00	0.446	0.602
Occupational	8	0.88 (0.49–1.56)	30.76	87.00	0.00	0.804	0.624
Environmental	6	0.87 (0.58–1.31)	31.96	90.61	0.00	0.567	0.497
Cancer severity ^a^							
19 articles							
Most aggressive type	4	1.23 (0.87–1.73)	24.7	91.9	0.00	0.652	0.602
11 articles							
Most aggressive type	3	1.50 (1.08–2.07)	6.68	70.07	0.035	0.971	0.602
8 articles							
Most aggressive type	1	1.00 (0.86–1.16)	/	/	/	/	/
NOS							
19 articles							
≥7	8	1.04 (0.97–1.11)	5.15	2.85	0.398	0.347	0.573
<7	15	1.26 (0.90–1.76)	344.05	96.51	0.00	0.826	0.020
11 articles							
≥7	/	/	/	/	/	/	/
<7	12	1.29 (0.75–2.21)	290.74	96.56	0.00	0.800	0.073
8 articles							
≥7	8	1.04 (0.97–1.11)	5.15	2.85	0.398	0.347	0.573
<7	3	0.86 (0.31–2.40)	28.01	96.43	0.00	/	/

^a^ Stratification by outcome (considered the most aggressive type). ^b^ Number of observations.

## Data Availability

The data presented in this study are available upon request from the corresponding author.

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
