# Peer review of "The Association Between Cadmium Exposure and Prostate Cancer: An Updated Systematic Review and Meta-Analysis"

_ijerph, 2024, doi:10.3390/ijerph21111532_

Round 1

Reviewer 1 Report

Comments and Suggestions for Authors

Critical review of the manuscript titled

”The Association of cadmium exposure and Prostate Cancer: An update Systematic Review and Meta-analysis”

submitted to the International Journal of Environmental Research and Public Health

The manuscript presents a systematic review and meta-analysis on the relationship between cadmium (Cd) exposure and prostate cancer morbidity. The topic is an important environmental health issue, although the results do not add much to the existing knowledge. The manuscript is written in acceptable English, although needs improvement at some places.

Major comments

1)      A general comment is that a meta-analysis is a possible addition to a systematic review; therefore, when referring to this study, it should be named systematic review and meta-analysis and not only a meta-analysis throughout the text.

2)      The methodology should include the PECO statement.

3)      The screening tool should be included in a supplementary file or at least named.

4)      The data extraction tool should be included in a supplementary file or at least named.

5)      It is better to apply the standard reasons for exclusion (e.g. wrong design, wrong population, wrong exposure, wrong outcome) in Figure 1.

6)      “The general characteristics of the 8 studies and relative populations included in the meta-analysis evaluating the association between Cd exposure and PCa are shown in Table 1.” (lines 199-201) – The 9th study that was not included in the meta-analysis is also in Table 1.

7)      In Table 1, there are 13 adopted studies indicated in red and 7 current studies. Please resolve this discrepancy. Probably Elghany NA, 1990 is a current finding.

8)      The new systematic review found two studies published years before the first systematic review (1995 and 2009). How were these studies missed by the first review?

9)      Why is the cut-off drawn at 7 for the 11 former studies, when their quality was assessed on a 6-scale?

10)  The hypothesis on the possible preventive effect of increased urinary Cd level is very weak. Increased Cd excretion can be a sign of kidney damage caused by long-term Cd exposure, which, if Cd causes prostate cancer, should be a sign of systemically increased and not decreased risk.

11)  The authors argue that Europeans are more exposed to Cd than people at other geographical locations. Since the study settings are different, a positive finding in European studies does not necessarily mean Europeans are more exposed. This argument could only be supported by a dose-response analysis. Western dietary pattern is also not a strong reasoning, since the same affluent diet is followed in North America, and actually by now at several places around the world.

Minor comments

1)      There are several minor errors of English, e.g.” ...higher Cd exposure is significantly associated with an increased of aggressive PCa...” (lines 21-22), “Accordingly…” (line 28.

2)      “Prostate cancer (PCa) is the most common cancer among men…” (line 11). It is not entirely true, since skin cancer is the most frequent.

3)      Prospective is not a study design (lines 84-85). Please rename it to cohort study.

4)      Replace “The full score was 7… (line 99) with “The full score was 9…”.

5)      The text in lines 100-103 is a duplicate.

6)      There is improper paragraph segmentation in the Statistical Analysis section.

7)      Some lines are not readable in Figure 1.

8)      Include the number of excluded studies during title/abstract screening in Figure 1.

9)      Sentence in lines 232-236 is unclear.

10)  Where is Table 1B (line 239)?

11)  Where is Table 3 (lines 359-360)?

Summary

The study is a useful update on the effect of Cd exposure on prostate cancer with two new findings. According to the results, increased urinary Cd may have preventive effect on prostate cancer. This observation deserves further studies. The other is that Cd exposure may be associated with more aggressive forms of prostate cancer, which should also be further studied.

Comments on the Quality of English Language

The manuscript is written in acceptable English, although needs improvement at some places (see in Comments and Suggestions).

Author Response

Major comments

1) A general comment is that a meta-analysis is a possible addition to a systematic review; therefore, when referring to this study, it should be named systematic review and meta-analysis and not only a meta-analysis throughout the text.

We thank the reviewer for pointing this out. We agree with this comment. Therefore, we have also added ‘systematic review and meta-analysis’ to ‘meta-analysis’ in the entire text

2)      The methodology should include the PECO statement.

We thank the reviewer for pointing this out. We agree with this comment. The PECO statement is reported in study protocol, but we have also added PECO statement in supplementary material (Table S1).

PECO statement

 Participants/population

Inclusion: People with prostate cancer from hospital and community (need to be diagnosed by a professional physician or histopathology).

Exclusion: People combined with other prostate diseases.

Intervention(s), exposure(s)

Cadmium exposure (Residential; Occupational; Dietary)

Comparator(s)/control

Subjects with higher than lower/no exposure of cadmium than exposure group

Main outcome(s)

The aim of this study is to examine the potential link between cadmium exposure and risk of prostate cancer in men by a systematic review of epidemiological studies and meta-analysis. The main outcomes are incidence prostate cancer.

3)      The screening tool should be included in a supplementary file or at least named.

We thank the reviewer for this advice but as depicted in the flowchart and described in lines 142-144, the initial screening process was performed on the titles and abstracts of the articles.

4)      The data extraction tool should be included in a supplementary file or at least named.

Thank you for your comment. As reported in section "2.3. Data Extraction and Quality Assessment" Information extracted from each selected study included: first author’s last name, publication year, country, study design, sample size, population characteristics (age, race, BMI, smoking status), follow-up duration for cohort studies, risk estimates with 95% CIs, Cd exposure details (biological sample, type of exposure, evaluation test), confirmation of PCa, and adjustment for confounding factors. Then, in Table 1 we reported the data extracted.

5)      It is better to apply the standard reasons for exclusion (e.g. wrong design, wrong population, wrong exposure, wrong outcome) in Figure 1.

We thank the reviewer for this suggestion. We have implemented your observation, and we have now applied the standard reasons for exclusion in Figure 1.

6)      “The general characteristics of the 8 studies and relative populations included in the meta-analysis evaluating the association between Cd exposure and PCa are shown in Table 1.” (lines 199-201) – The 9th study that was not included in the meta-analysis is also in Table 1.

We thank the reviewer for this advice. We better explained the as follows:

 "The general characteristics of the 9 studies and relative populations included in the systematic review evaluating the association between Cd exposure and PCa are shown in Table 1. Eight of these studies are included in the systematic review and meta-analysis [36-43]."

7)      In Table 1, there are 13 adopted studies indicated in red and 7 current studies. Please resolve this discrepancy. Probably Elghany NA, 1990 is a current finding.

We thank the reviewer for pointing this out. We have revised Table 1 and corrected the discrepancy in the previous version of the manuscript. The 9 studies included in black are those that are included in our new systematic review and meta-analysis. There was a typo, Aronson KJ and Elghany NA are included in our new systematic review and meta-analysis. This confirms the 9 studies included in our new systematic review and meta-analysis and the 11 studies included in the previous systematic review and meta-analysis

8)      The new systematic review found two studies published years before the first systematic review (1995 and 2009). How were these studies missed by the first review?

We agree with the reviewer’s assessment but there is indeed a discrepancy in the search databases used. In the previous meta-analysis, the authors used the Embase and PubMed databases, while we used PubMed, Scopus, and Web of Science.

9)      Why is the cut-off drawn at 7 for the 11 former studies, when their quality was assessed on a 6-  scale?

Thank you for your comment. The first systematic review and meta-analysis decided to use a quality score greater than 5. In our new systematic review and meta-analysis, to increase the accuracy of our findings, we chose to set the minimum quality level of our studies at 7, as recommended by several studies.

(https://journals.sagepub.com/doi/10.1177/03946320241296903, https://www.sciencedirect.com/science/article/pii/S1036731423000875, https://www.sciencedirect.com/science/article/pii/S0020748923001670)

10)  The hypothesis on the possible preventive effect of increased urinary Cd level is very weak. Increased Cd excretion can be a sign of kidney damage caused by long-term Cd exposure, which, if Cd causes prostate cancer, should be a sign of systemically increased and not decreased risk.

We thank the reviewer for this advice, giving us an interesting alternative possible explanation of the risk assessment in those with increased urinary Cd level. We discussed this important aspect in discussion section as follows:

“Anyway, this is only an hypothesis that need further studies since we can not exclude that increased Cd excretion can be a sign of kidney damage caused by long-term Cd exposure, which could be a sign of systemically increased and not decreased risk”

11)  The authors argue that Europeans are more exposed to Cd than people at other geographical locations. Since the study settings are different, a positive finding in European studies does not necessarily mean Europeans are more exposed. This argument could only be supported by a dose-response analysis. Western dietary pattern is also not a strong reasoning, since the same affluent diet is followed in North America, and actually by now at several places around the world.

We thank the reviewer for this consideration, and we agree on the difficulty of assessing the exact exposure to Cd among different countries.  This is why in discussion section we further explained this concept as follows:

“Nonetheless, it is important to highlight that, although we found positive findings in European studies, we did not perform a dose-response analysis, as this was beyond the scope of the work. Therefore, we can only hypothesize that Europeans may have higher cadmium (Cd) exposure, though we lack direct evidence. Additionally, other factors beyond dietary patterns should be considered with further investigation”.

Minor comments

1)There are several minor errors of English, e.g.” ...higher Cd exposure is significantly associated with an increased of aggressive PCa...” (lines 21-22), “Accordingly…” (line 28.

We thank the reviewer for pointing this out. We have corrected

2)       “Prostate cancer (PCa) is the most common cancer among men…” (line 11). It is not entirely true, since skin cancer is the most frequent.

Thank you for this suggestion. We have corrected with “Prostate cancer (PCa) is one of common cancer among men”.

3)      Prospective is not a study design (lines 84-85). Please rename it to cohort study.

We thank the reviewer for pointing this out. We have corrected with “cohort study”.

4)      Replace “The full score was 7… (line 99) with “The full score was 9…”.

We thank the reviewer for this advice. As suggested by the reviewer, we have corrected the previous full score with the correct one.

5)      The text in lines 100-103 is a duplicate.

We carefully checked in lines suggested by reviewer, and we removed possible duplicate sentences.

6)      There is improper paragraph segmentation in the Statistical Analysis section.

We followed reviewer's advice and we have corrected the improper paragraph segmentation in the Statistical Analysis section

7)      Some lines are not readable in Figure 1.

We apologize for the error and have replaced Figure 1 with a new version that is more readable and clearer.

8)      Include the number of excluded studies during title/abstract screening in Figure 1.

Thank you for pointing this out. We apologize for the error. However, the previous Figure 1 did not clearly present the number of articles excluded during the title/abstract screening. With the updated Figure 1, the requested data is now accessible.

9)      Sentence in lines 232-236 is unclear.

We agree with the reviewer and we apologize for this. To increase clarity of our sentence we reformulated the text as follows:

Case-control studies show an increased likelihood of PCa risk of OR 1.17 (95% CI: 0.64-2.14), not statistically significant, with a high heterogeneity (Q = 186.37 and I²=93.56%). Regarding the subset including the latter 11 articles, cohort studies have an increased likelihood of PCa risk OR of 1.13 (95% CI: 1.03-1.23), statistically significant. This result showed no heterogeneity (Q = 0.07 and I² = 0.00%). On the other hand, case-control studies of the same subset of 11 latter articles, did not show a statistically significant increased likelihood of PCa and Cd exposure”.

10)  Where is Table 1B (line 239)?

We thank the reviewer for this advice but we were referring to Table 1. It was a typo there is no Table 1A and 1B.

11)  Where is Table 3 (lines 359-360)?

We thank the reviewer for this advice but we were referring to Table 1. It was a typo, we meant Table 2 and Figure 3.

Reviewer 2 Report

Comments and Suggestions for Authors

The aim of the study was to update previous studies. The manuscript provides a good consolidation of both recent and past data. The authors confirm earlier findings regarding the lack of a significant association between cadmium levels and prostate cancer, considering the overall effect. A new aspect of this analysis was the stratification by biological sample type and cancer severity. The authors note some limitations of the study but also recognize the need for future research to address the unclear connection between Cd and PCa. The manuscript is generally well-written and well-structured.  The authors employed a range of statistical methods, and results are well-discussed and conclusive. However there are a few mistakes and issues:

1) line 75 “((Cadmium..” too many brackets

2) In the “Introduction” the authors should provide more information about cadmium, specifically which types of cancer are it is linked to and include more data on the incidence of PCa in different populations (e.g., African vs. European)

3) In the ”Results”: Figure 1 - I think that the section “Included” appears too short (possibly cut off)

4) line 176 – “in the” table

5) lines 187-188 – The sentence is unclear and may contain an error

6) line 220 – examining the type of biological sample – authors should explain what connections were studied, in this form it is too short to understand it clearly.

7) line 231 – slight association – please define the association with … (what?) and so on in the whole text. It is better for the readers to indicate it clearly.

8) line 334 – too much dots

Comments on the Quality of English Language

Minor editing of English language required.

Author Response

1) line 75 “((Cadmium..” too many brackets

We thank the reviewer for this advice and we have corrected in the manuscript.

2) In the “Introduction” the authors should provide more information about cadmium, specifically which types of cancer are it is linked to and include more data on the incidence of PCa in different populations (e.g., African vs. European) (da insierire)

We agreed with the reviewer to improve the introduction and we added the following information:

“Cadmium (Cd) is classified as a Group 1 human carcinogen by the IARC and has been linked to several types of cancers, including lung, kidney, bladder, and prostate cancers  [1-2]. Specifically, Cd exposure has been implicated in the development of more aggressive and advanced stages of prostate cancer (PCa), making it a significant concern for public health. Additionally, the incidence of PCa shows considerable variation across different populations. For instance, African-American men, have higher rates of PCa incidence and mortality compared to European men, with genetic, environmental, and socio-economic factors that playing a role. To understand the relationship between Cd exposure and these disparities in PCa incidence is crucial for targeted prevention strategies [3].”

[1]        IARC Working Group on the Evaluation of Carcinogenic Risks to Humans. Arsenic, Metals, Fibres and Dusts. Lyon (FR): International Agency for Research on Cancer; 2012. (IARC Monographs on the Evaluation of Carcinogenic Risks to Humans, No. 100C.) CADMIUM AND CADMIUM COMPOUNDS. Available from: https://www.ncbi.nlm.nih.gov/books/NBK304372/

[2]        “Cadmium and Cadmium Compounds (IARC Summary & Evaluation, Volume 58, 1993).” Accessed: Oct. 24, 2024. [Online]. Available: https://www.inchem.org/documents/iarc/vol58/mono58-2.html

[3]        W. Rayford et al., “Comparative analysis of 1152 African-American and European-American men with prostate cancer identifies distinct genomic and immunological differences,” Communications Biology 2021 4:1, vol. 4, no. 1, pp. 1–9, Jun. 2021, doi: 10.1038/s42003-021-02140-y.

3) In the ”Results”: Figure 1 - I think that the section “Included” appears too short (possibly cut off)

We thank the reviewer for pointing this out. We apologize for the error and have replaced Figure 1 with a new version that is more readable and clearer.

4) line 176 – “in the” table

We thank the reviewer for this advice and we corrected it.

5) lines 187-188 – The sentence is unclear and may contain an error

We agree with reviewer to better clarify this aspect and we added in the text the following sentence:

Armstrong & Kazantzis assessed exposure using occupational history. Checkoway et al. conducted a study with 40 cases and 40 controls, utilizing a structured occupational questionnaire, similar to the approach taken by van der Gulden et al.

6) line 220 – examining the type of biological sample – authors should explain what connections were studied, in this form it is too short to understand it clearly. (da inserire)

We thank the reviewer for this advice. We changed the new sentence reads as follows:

“When examining the type of biological sample, we analyzed the relationship between Cd concentrations in blood and urine and their association with PCa risk. Specifically, we assessed how Cd levels in these samples correlate with the risk of developing PCa. Data from 19 studies showed that blood Cd levels presented an OR of 0.75 (95% CI: 0.51–1.10), while urine Cd levels yielded an OR of 0.74 (95% CI: 0.56–0.98).”

7) line 231 – slight association – please define the association with … (what?) and so on in the whole text. It is better for the readers to indicate it clearly.

We appreciate the reviewer's advice and we always clearly stated the type of association

8) line 334 – too much dots

We agree with the reviewer and we removed extra dots.

Reviewer 3 Report

Comments and Suggestions for Authors

This is an interesting paper focusing on cadmium effect on PCa development and aggressiveness. The meta-nalysis is an update of previous studies.

The results are in my opinion the basis for future researchs.

Authors need to clarify the strengths compared previous meta-analyses.

Author Response

This is an interesting paper focusing on cadmium effect on PCa development and aggressiveness. The meta-analysis is an update of previous studies.

The results are in my opinion the basis for future researchs.

Authors need to clarify the strengths compared previous meta-analyses.

We thank the reviewer to appreciate our work. We improved the strengths paragraph highlighting differences with previous meta-analyses, focusing primarily on prostate cancer severity. This has allowed us to provide how increased cadmium exposure may be linked to the development of more aggressive PCa.

In conclusion, while the updated systematic review and meta-analysis reinforces the findings of the previous study, it also provides new insights through further stratifications analyses and considerations. Specifically, this updated systematic review and meta-analysis integrates recent studies that reinforce the actual evidences but also promote future experimental research. A critical focus of this updated systematic review and meta-analysis is on disease severity, where findings suggest a strong association between Cd exposure and the development of more aggressive forms of PCa. These insights represent an important strength of our work and if validated by future research they could be useful in developing targeted public health interventions aimed at reducing Cd exposure. Further studies are necessary to explore these associations more thoroughly.

Round 2

Reviewer 1 Report

Comments and Suggestions for Authors

Critical review of the revised manuscript titled

”The Association of cadmium exposure and Prostate Cancer: An update Systematic Review and Meta-analysis”

submitted to the International Journal of Environmental Research and Public Health

The manuscript presents a systematic review and meta-analysis on the relationship between cadmium (Cd) exposure and prostate cancer morbidity. The topic is an important environmental health issue, although the results do not add much to the existing knowledge. The manuscript is written in acceptable English, although needs improvement at some places.

Major comments

1)      Prostate cancer is the outcome of the investigated association (PECO outcome is not the aim of the study) and the population is human population. The PECO statement should be rephrased accordingly.

2)      Regarding the screening tool, the original question was about the tool/platform of the screening and not about the process or the results. Was the screening done e.g. using Excel or an online systematic review platform?

3)      Regarding the extraction tool, the original question was targeting the same information as with the screening tool. Was the data extraction done e.g. using Excel or an online systematic review platform?

Minor comments

1)     There are still several minor errors of English, which need correction.

Summary

The manuscript is substantially improved but there are three major methodological points that still need clarification.

Comments on the Quality of English Language

  There are still several minor errors of English, which need correction.

Author Response

REVIEWER 1 ROUND 2

Major comments

1) Prostate cancer is the outcome of the investigated association (PECO outcome is not the aim of the study) and the population is human population. The PECO statement should be rephrased accordingly.

We thank the reviewer for this precious comment we changed our PECO statement accordingly as follows:

“Participants/population

Inclusion: Human male population from hospital and community.

Exclusion: People combined with other prostate diseases.

Exposure(s)

Cadmium exposure (Residential; Occupational; Dietary)

Comparator(s)/control

Subjects with higher than lower/no exposure of cadmium than exposure group

Main outcome(s)

Prostate cancer”

2) Regarding the screening tool, the original question was about the tool/platform of the screening and not about the process or the results. Was the screening done e.g. using Excel or an online systematic review platform?

We apologize for not having fully understood the meaning of your initial observation. We added this information in Search Strategy and Data source as follows:

“All results were then screened with Microsoft® Excel® for Microsoft 365 MSO (Version 2409 Build 16.0.18025.20160) 32 bit version”

3) Regarding the extraction tool, the original question was targeting the same information as with the screening tool. Was the data extraction done e.g. using Excel or an online systematic review platform?

Again, we truly apologize for the misunderstanding. We added this information in Data Extraction and Quality Assessment  paragraph as follows:

“Data extraction have been obtained through Microsoft® Excel® for Microsoft 365 MSO (Version 2409 Build 16.0.18025.20160) 32-bit version and then added to a table in Microsoft® Word for Microsoft 365 MSO (Version 2409 Build 16.0.18025.20160) 32-bit version” 

Minor comments

1)     There are still several minor errors of English, which need correction.

We thank the reviewer for this advice. We carefully checked the article, removing possible typos and/or gramma errors.